# CAT: COLLABORATIVE ADVERSARIAL TRAINING

## ABSTRACT

Adversarial training can improve the robustness of neural networks. Previous adversarial training methods focus on a single training strategy and do not consider the collaboration between different training strategies. In this paper, we find different adversarial training methods have distinct robustness for sample instances. For example, an instance can be correctly classified by a model trained using standard adversarial training (AT) but not by a model trained using TRADES, and vice versa. Based on this phenomenon, we propose a collaborative adversarial training framework to improve the robustness of neural networks. Specifically, we simultaneously use different adversarial training methods to train two robust models from scratch. We input the adversarial examples generated by each network to the peer network and use the peer network's logit to guide its network's training. **C**ollaborative **A**dversarial **T**raining (**CAT**) can improve both robustness and accuracy. Finally, Extensive experiments on CIFAR-10 and CIFAR-100 validated the effectiveness of our method. CAT achieved new state-of-the-art robustness without using any additional data on CIFAR-10 under the Auto-Attack benchmark[1].

## 1 INTRODUCTION

With the development of deep learning, Deep Neural Networks (DNNs) have been applied to various fields, such as image classification (He et al., 2016), object detection (Redmon et al., 2016), semantic segmentation (Pal & Pal, 1993), etc. And state-of-the-art performance has been obtained. But recent research has found that DNNs are vulnerable to adversarial perturbations (Goodfellow et al., 2014). A finely crafted adversarial perturbation by a malicious agent can easily fool the neural network. This raises security concerns about the deployment of neural networks in security-critical areas such as Autonomous driving (Chen et al., 2019) and medical diagnostics (Kong et al., 2017).

To cope with the vulnerability of DNNs, different types of methods have been proposed to improve the robustness of neural networks, including adversarial training (Madry et al., 2017), defensive distillation (Papernot et al., 2016), feature denoising (Xie et al., 2019) and model pruning (Madaan et al., 2020). Among them, Adversarial Training (AT) is the most effective method to improve adversarial robustness. AT can be regarded as a type of data augmentation strategy that trains neural networks based on adversarial examples crafted from natural examples. AT is usually formulated as a min-maximization problem, where the inner maximization generates adversarial examples, while the outer minimization optimizes the parameters of the model based on the adversarial examples generated by the inner maximization process.

The previous methods have focused on how to improve the model's adversarial accuracy, focusing only on the numerical improvement, but not on the characteristics of the different methods. We ask: *Do different adversarial training methods perform the same for different sample instances?* We analyzed different adversarial training methods (taking AT (Madry et al., 2017) and TRADES (Zhang et al., 2019) as examples) and found that different methods behave differently for sample instances, as illustrated in Figure 1. Specifically, for the same adversarial example, the network trained by AT can classify correctly, while the network trained by TRADES misclassifies. Similarly, some examples can be correctly classified by the network trained by TRADES, but not by the network trained by AT. That is, although AT and TRADES have the same numerical adversarial accuracy, they behave differently for sample instances. This raises the question:

*Do two networks learn better if they collaborate?*

---

[1] https://github.com/fra31/auto-attack

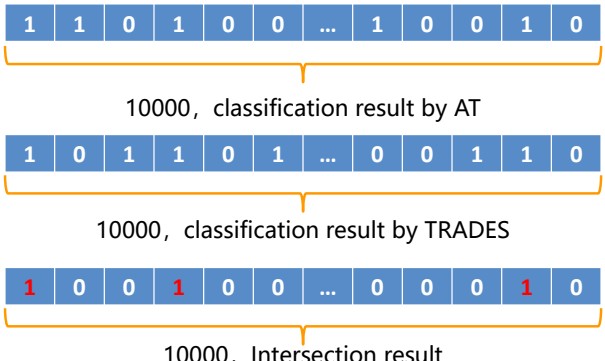

Figure 1: Classification results of different adversarial training methods on sample instances. The first row is the classification result of the model trained by AT and the second row is the classification result of the model trained by TRADES. 1 means correct classification and 0 means incorrect classification. 10000 is the size of the CIFAR-10 test set. The third row is the result of correct classification by both AT and TRADES, and the result is shown in red. It can be seen that the models trained by different methods perform differently on sample instances.

Based on this observation, we propose a **C**ollaborative **A**dversarial **T**raining (**CAT**) framework to improve the robustness of neural networks. Our framework is shown in Figure 2. Specifically, we simultaneously train two deep neural networks separately using different adversarial training methods. At the same time, the adversarial examples generated by this network are input to the peer network to obtain the corresponding logit. Then the logit obtained by the peer network is used to guide the learning of this network together with its own adversarial training objective function. We expect to improve the robustness of the neural network by allowing peers to learn from each other in this collaborative learning way. Extensive experiments on different neural networks and different datasets demonstrate the effectiveness of our approach. CAT achieved new state-of-the-art robustness without any additional synthetic data or real data on CIFAR-10 under the Auto-Attack benchmark.

In summary, our contribution is threefold as follows.

- We find that the models obtained using different adversarial training methods have different representations for individual sample instances.
- We propose a novel adversarial training framework: collaborative adversarial training. CAT simultaneously trains two neural networks from scratch using different adversarial training methods and allows them to collaborate to improve the robustness of the model.
- We have conducted extensive experiments on a variety of datasets and networks, and evaluated them on state-of-the-art attacks. We demonstrate that CAT can substantially improve the robustness of neural networks and obtain new state-of-the-art performance without any additional data.

## 2 RELATED WORK

### 2.1 ADVERSARIAL ATTACK

Since Goodfellow et al. (2014) discovered that DNNs are vulnerable to adversarial examples, a large number of works have been proposed to craft adversarial examples. Based on the accessibility to the knowledge of the target model, it can be divided into white-box attacks and black-box attacks. White-box attacks craft adversarial examples based on the knowledge of the target model, while black-box attacks are agnostic to the knowledge of the target model.

**White-box Attack:** Goodfellow et al. (2014) proposed FGSM to efficiently craft adversarial examples, which can be generated in just one step. Later FGSM was extended to different iterative

attack methods. I-FGSM performs FGSM iteratively with a small step size. Madry et al. (2017) proposed PGD to generate adversarial examples, which is the most efficient way of using the first-order information of the network. Dong et al. (2018) combines momentum into the iterative process to help the model escape from local optimal points. And the adversarial examples generated by this method are also more transferable. Boundary-based attacks such as deepfool (Moosavi-Dezfooli et al., 2016) and CW (Carlini & Wagner, 2017) also make the model more challenging. Recently, the ensemble approach of diverse attack methods (Auto-Attack), consisting of APGD-CE (Croce & Hein, 2020b), APGD-DLR (Croce & Hein, 2020b), FAB (Croce & Hein, 2020a) and Square Attack (Andriushchenko et al., 2020), became a benchmark for testing model robustness.

**Black-box Attack:** Due to the similarity of the model structure, the adversarial examples generated on the surrogate model can be transferred to fool the target model. There are many ways to explore the transferability of adversarial examples for black-box attacks. Dong et al. (2018) combines momentum with an iterative approach to obtain better transferability. Scale-invariance (Lin et al., 2019) boosts the transferability of adversarial examples by transforming the inputs on multiple scales.

## 2.2 ADVERSARIAL ROBUSTNESS

Adversarial attacks present a significant threat to DNNs. For this reason, many methods have been proposed to defend against adversarial examples, including denoising (Xie et al., 2019), adversarial training (Madry et al., 2017), data aumentation (Rebuffi et al., 2021), and input purification (Naseer et al., 2020). ANP (Madaan et al., 2020) finds the vulnerability of latent features and uses pruning to improve robustness. Madry uses PGD to generate adversarial examples for adversarial training, which is also the most effective way to defend against adversarial examples. A large body of work uses new regularization or objective functions to improve the effectiveness of standard adversarial training. Kannan et al. (2018) uses Adversarial logit pairing to improve robustness by encouraging the logits of normal and adversarial examples to be closer together. TRADES (Zhang et al., 2019) uses KL divergence to regularize the output of adversarial and pure examples.

## 2.3 KNOWLEDGE DISTILLATION

Knowledge distillation (KD) is commonly used for model compression and was first used by hinton (Hinton et al., 2015) to distill knowledge from a well-trained teacher network to a student network. KD can significantly improve the accuracy of student models. There have been many later works to improve the effectiveness of KD (Romero et al., 2014). In recent years, KD has been extended to other areas. Goldblum et al. (2020) analyzes the application of knowledge distillation to adversarial robustness and proposes ARD to transfer knowledge from a large teacher model with better robustness to a small student model. ARD can produce a student network with better robustness than training from scratch. In this paper, we propose a more effective collaborative training framework to improve the robustness of the network.

## 3 PROPOSED METHOD

### 3.1 MOTIVATION

We investigated the performance of the robust models obtained by different training methods on sample instances. We found that different models perform differently on sample instances: for some samples, the model trained by correctly classifies AT (Madry et al., 2017), while the model trained by TRADES (Zhang et al., 2019) misclassifies. Similarly, for some samples, the network trained by TRADES can classify correctly, while the network trained by AT can misclassify. A straightforward conclusion can be drawn that the networks trained by different methods master different knowledge, although their accuracy values are about the same. So can we use the knowledge learned from these two networks to improve the robustness of neural networks? A simple idea is to let two networks that master different knowledge learn collaboratively. For this purpose, we propose collaborative adversarial training. CAT improves the robustness of neural networks by making the knowledge of both networks interact during the training procedure. And the framework is illustrated in Figure 2.

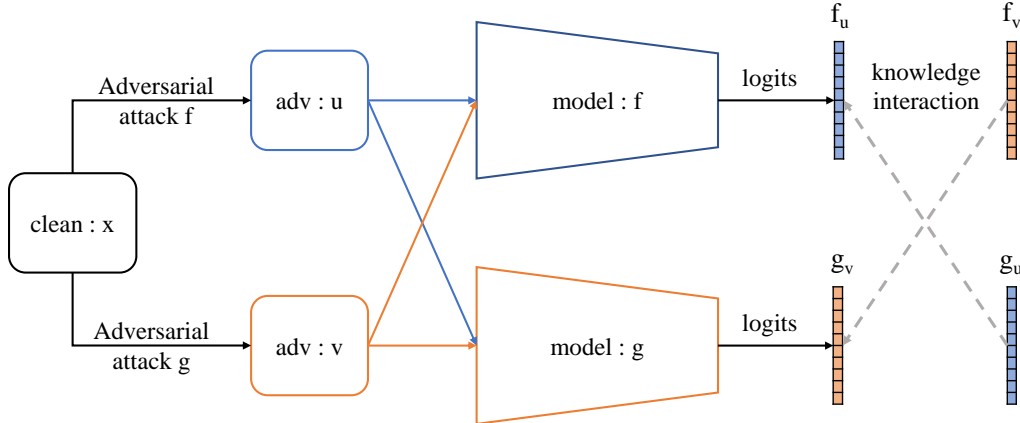

Figure 2: **The framework of CAT, performing adversarial training collaboratively.** Given a batch of natural samples, the two networks $f$ and $g$ are attacked separately to generate adversarial examples u and v. Then u and v are fed into both networks to obtain the corresponding logits. We then use the logits obtained from the peer networks to guide the learning of its network, i.e., $g_u \to f_u, f_v \to g_v$.

## 3.2 COLLABORATIVE ADVERSARIAL TRAINING (CAT)

We take the methods of AT and TRADES as an example to introduce collaborative adversarial training. We first briefly introduce the training objective functions of AT and TRADES and then introduce CAT in detail.

Adversarial training is defined as a min-maximization problem, and Madry et al. (2017) proposes to use PGD for adversarial training. That is, PGD is used to generate adversarial examples for the internal maximization process, while external minimization uses the internal PGD-generated adversarial examples and the ground-truth label $y$ to optimize the model parameters. AT is formulated as:

$$\min_{\theta} \mathbb{E}_{(x,y)\in D_{data}}(\arg\max_{\delta} L(f_{\theta}^{AT}(x_{AT}^{adv}), y)), \tag{1}$$

$$x_{AT}^{adv} = x + \delta. \tag{2}$$

where $D_{data}$ is the training data distribution, $x$ and $y$ are training data and corresponding label samples from $D_{data}$. $f_{\theta}$ is a neural network parameterized by $\theta$. $L$ is the standard cross-entropy loss used in image classification tasks. $\delta$ is the adversarial perturbations generated by PGD. Follwing previous study, $\delta$ is bounded by $l_{\infty}$.

Neural Networks trained by AT can obtain a certain level of robustness, with compromises on the accuracy of natural samples. For this purpose, TRADES uses a new training objective function for adversarial training. Formulated as:

$$\min_{\theta'} \mathbb{E}_{(x,y)\in D_{data}} L(g_{\theta'}^{TRADES}(x), y) + \lambda D_{KL}(g_{\theta'}^{TRADES}(x), g_{\theta'}^{TRADES}(x_{TRADES}^{adv})), \tag{3}$$

where $x^{adv}$ is the adversarial data corresponding to natural data $x$ and y is the true label. $L$ is the cross-entropy loss in classification task. $D_{KL}$ is the KL divergence to pushing natural logits and adversarial logits together. $\lambda$ is a trade-off parameter.

CAT expects to improve robustness by letting neural networks trained by different methods exchange knowledge information, i.e., collaborative adversarial learning. As illustrated in Figure 2. we use the logit of a peer network to guide the learning of this network. Specifically, we input the adversarial data crafted by the network trained by AT into the network trained by TRADES to get the corresponding logit. Then use the logit obtained by the network trained by TRADES to guide the training of the network trained by AT. The formulation goes to:

$$L_1 = D_{KL}(f^{AT}(x_{AT}^{adv}), \hat{g}^{TRADES}(x_{AT}^{adv})), \tag{4}$$

where $D_{KL}$ is KL divergence used to compute the relative entropy, the same as in TRADES. $f^{AT}$ is the network trained with AT and $g^{TRADES}$ is the network trained with TRADES. $\hat{g}^{TRADES}(x_{AT}^{adv})$ represents that we take the logit obtained by network trained by TRADES as a constant. $x_{AT}^{adv}$ is the adversarial data generated by $f^{AT}$ with PGD function based on natural example $x$.

Similarly, to make the two networks learn collaboratively. We need to feed the adversarial samples generated by the TRADES network to the AT network to obtain the corresponding logit. And then the logit obtained by the peer network is used to guide the training of the network trained by TRADES. The loss is formulated as:

$$L_2 = D_{KL}(g^{TRADES}(x_{TRADES}^{adv}), \hat{f}^{AT}(x_{TRADES}^{adv})). \qquad (5)$$

$x_{TRADES}^{adv}$ is the adversarial example crafted by network trained by TRADES use KL divergence function. $\hat{f}^{AT}(x_{TRADES}^{adv})$ represents that we take the logit obtained by network trained by AT as a constant.

It is not enough to let the two networks learn from each other in this way. Real class labels are needed to guide them. For this purpose, we combine the respective training objective functions of the two networks and the mutual learning objective function to guide the learning of the networks together. Therefore, the training objective function for collaborative adversarial training based on AT and TRADES is:

$$L_{total} = \alpha L_{TRDES} + (1 - \alpha)L_2 + \beta L_{AT} + (1 - \beta)L_1, \qquad (6)$$

where $\alpha$ and $\beta$ is the trade-off parameter to balance the guidance of peer logit and the original training objective function. $L_{TRADES}$ is the training objective of TRADES defined in Equation (3). And $L_{AT}$ is the training objective of AT defined in Equation (1). The first two items in Equation (6) are used to train model $g$ and the last two items are used to train model $f$, due to we take the peer logit as constant.

The decision boundaries learned by different adversarial training methods are different. Under the constraint of peer logit, i.e., Equation (4) and Equation (5), the two networks trained by different methods continuously optimize the classification decision boundaries in the process of collaborative learning. Finally, both networks learn better decision boundaries than learning alone to obtain better robustness.

Our collaborative adversarial learning is a generalized adversarial training method that can be used with any two adversarial methods. Generally, CAT can use any number of different adversarial training methods for collaborative learning. Results of CAT with three adversarial training methods are delayed to Appendix.

## 4 EXPERIMENT RESULTS

In this section, we conduct extensive experiments on popular benchmark datasets to demonstrate the effectiveness and performance of CAT. First, we briefly introduce the experiment setup and implementation details of CAT. Then, ablation studies are done to choose the best hyperparameters and CAT methods. Finally, according to the best CAT methods, we report the white-box and black-box adversarial robustness on two popular benchmark datasets. All images values are scaled into [0,1], and all our experiments are run on a single NVIDIA GeForce GTX 1080Ti.

**Datasets:** We used two benchmark datasets, including CIFAR-10 (Krizhevsky et al., 2009) and CIFAR-100 (Krizhevsky et al., 2012). CIFAR-10 has 10 classes. For each class, there are 5000 images for training and 1000 images for test. And for CIFAR-100, there are 100 classes, and similarly for each class, there are 500 images for training and 100 images for test. Both datasets are widely used for training and testing adversarial robustness. The image size in both datasets is $32 \times 32$.

**Training setup:** Our overall training parameters refer to Madaan et al. (2020). Specifically, we use SGD (momentum 0.9, batch size 128) to train ResNet18 for 200 epochs on the CIFAR-10 dataset with weight decay 5e-4 and initial learning rate 0.1 which is divided by 10 at 100-th and 150-th epoch, respectively. RandomCrop and RandomHorizontalFlip are used as data augmentation. For the internal maximization process, we use $PGD_{10}$ adversarial attack to solve, with random start, step size 2.0/255, and perturbation size 8.0/255. We use $\lambda = 100$ in all experiments. The experimental

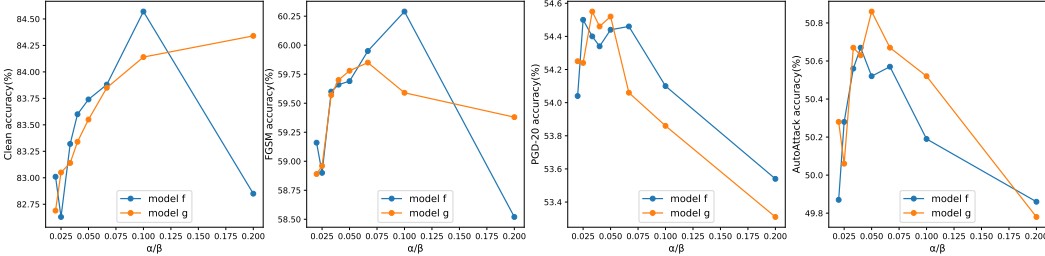

Figure 3: Adversarial robustness using different hyperparameters under the trades-at for collaborative adversarial training framework. From left to right, the results of Clean acc, FGSM acc, PGD acc, and AA acc are shown. model $f$ and model $g$ represent the results of using TRADES and AT in the CAT training framework, respectively.

Table 1: The white-box robustness results (accuracy (%)) of different CAT methods on CIFAR-10. We report the results of best checkpoint and last checkpoint. The best results are marked using **boldface**. Two ResNet-18 networks are used in our CAT framework.

| Method | Best Checkpoint | | | | | Last Checkpoint | | | | |
|---|---|---|---|---|---|---|---|---|---|---|
| | Clean | FGSM | PGD$_{20}$ | CW$_\infty$ | AA | Clean | FGSM | PGD$_{20}$ | CW$_\infty$ | AA |
| CAT$_{AT-TRADES}$ | 83.74 | 59.69 | 54.44 | 52.60 | 50.52 | 84.45 | **60.03** | **53.01** | **52.01** | 49.30 |
| | 83.55 | 59.78 | **54.52** | 52.58 | 50.86 | 84.12 | 59.69 | 52.82 | 51.88 | 49.39 |
| CAT$_{AT-ALP}$ | 84.66 | 59.94 | 53.11 | 51.90 | 49.74 | 84.71 | 59.84 | 50.77 | 50.53 | 47.80 |
| | **85.21** | **60.21** | 53.02 | 52.13 | 49.96 | **85.27** | 59.75 | 51.10 | 50.69 | 47.91 |
| CAT$_{TRADES-ALP}$ | 83.91 | 59.76 | 54.44 | 52.56 | **51.02** | 84.67 | 59.85 | 52.51 | 51.43 | 49.31 |
| | 84.75 | 59.76 | 54.17 | **52.72** | 50.85 | **85.27** | 59.82 | 52.56 | 51.83 | **49.64** |

parameters of ResNet18 in CIFAR-100, WideResNet-34-10 in CIFAR-10 and CIFAR-100 are the same as described above.

**Evaluation setup:** We report the clean accuracy on natural examples and the adversarial accuracy on adversarial examples. For adversarial accuracy, we report both white-box and black-box. We follow the widely used protocols in the adversarial research field. For the white-box attack, we consider three basic attack methods: FGSM (Goodfellow et al., 2014), PGD (Madry et al., 2017), and CW$_\infty$ (Carlini & Wagner, 2017) optimized by PGD$_{20}$, and a stronger ensemble attack method named AutoAttack (AA) (Croce & Hein, 2020b). For the black-box attacks, we consider both transfer-based attacks and query-based attacks.

Due to the fact that CAT uses two methods for collaborative training, we report the results for both networks. For example, the method of collaborative adversarial training using TRADES and ALP can be denoted as CAT$_{TRADES-ALP}$.

## 4.1 ABLATION STUDY

### 4.1.1 HYPERPARAMETER:

CAT improves adversarial robustness through the learning of collaboration, which requires both the knowledge of peer networks and the guidance of the ground truth label. The balance of original methods and the collaborative is a trade-off by a hyperparameter $\alpha$ and $\beta$. For simplicity, we set $\alpha$ equals $\beta$ in our experiment. We execute collaborative training by TRADES and AT as the base method and experiment with different trades-off parameters. We test various $\alpha$ values varying from 1/50 to 1/5. The experiment results are illustrated in Figure 3. From the figure we can conclude that if $\alpha$ is too high, i.e., little knowledge is drawn from the peer network, the effect is about the same as training with AT and trades alone. If $\alpha$ is too low, i.e., overly focused on the knowledge from the

peer network, The network is also not very robust. Since Auto-Attack is currently the most powerful integrated attack method, we choose hyperparameters $\alpha$ and $\beta$ based primarily on the robustness of the network against AA. In the following experiments, we set $\alpha = \beta = 1/20.0$ by default.

### 4.1.2 DIFFERENT CAT METHODS:

As described in Section 3.2, any two adversarial training methods can be incorporated into the CAT framework and learned collaboratively. Considering that different adversarial training methods have distinct properties, the performance of different CAT methods may also vary. For this reason, we consider three collaborative adversarial training methods, AT-TRADES, AT-ALP, and TRADES-ALP, respectively. Table 1 shows the performance of the different CAT methods. All CAT training methods obtained good robustness against four attack methods. We again mainly consider the performance of auto-attack and choose TRADES-ALP as the base method for our CAT.

## 4.2 ADVERSARIAL ROBUSTNESS

### 4.2.1 WHITE-BOX ROBUSTNESS

For FGSM, PGD, $CW_\infty$, AA, the attack perturbations are all 8.0/255 and the step size for PGD, $CW_\infty$ are 2/25, with 20 iterations. According to the way of reporting in previous papers, we report both the best checkpoint and the last checkpoint of the training phase. The best checkpoint result of the training phase is selected based on the model's PGD defense results for the test dataset (attack step size 2.0/255, iteration number 10, perturbation size 8.0/255).

Table 2 shows the adversarial accuracy of the networks trained by different methods on two datasets CIFAR-10 and CIFAR-100 against the four attacks. We also report the accuracy of the model for natural examples. From the table, we can obtain the following conclusions: (1) our method obtains good robustness against all four attacks on both datasets. For example, for the strongest AA attack method, CAT can obtain 2% improvement. (2) our method obtains high adversarial robustness while ensuring accuracy for natural examples. Although there is still a big gap compared to 94.65 of the standard training strategy, there is a nearly 1% improvement in the accuracy of the natural examples compared to the other three methods. (3) The robustness of both networks is significantly improved in the CAT training framework. The robustness of both networks is higher than separately trained ones. (4) The difference in accuracy between the two networks trained in the CAT framework is smaller than separately trained ones, which demonstrates that the two networks do well in collaborative learning. For example, for the CIFAR-10 dataset, the difference in robustness between the two networks of TRADES-ALP against AA in the CAT training framework is 0.17%, while the difference is 0.73% under separate training. The same conclusion can be drawn from CIFAR-100 datasets.

### 4.2.2 BLACK-BOX ROBUSTNESS

For black-box attacks, we consider both transfer-based attacks and query-based attacks. For the transfer-based attack, we use the standard adversarial training of ResNet-34 as the surrogate model, trained with the same parameters as described in Section 4. First, we perform the attack on the surrogate model to generate adversarial examples and then transfer the adversarial examples to the target network to get the robustness of the target network. Here, we consider four attacks: FGSM, $PGD_{20}$, $PGD_{40}$, and $CW_\infty$, with the same attack parameters as Section 4.2.1. For query-based attacks, we consider Square attack, which is an efficient black-box query-based attack method. Table Table 3 shows the results, and our method CAT achieves the best performance.

## 4.3 COMPARISION TO SOTA

We use two WideResNet-34-10 (Zagoruyko & Komodakis, 2016) networks for collaborative adversarial training, one using the TRADES (Zhang et al., 2019) training method and the other using the ALP (Kannan et al., 2018) training method. Table 4 shows the accuracy of the different methods for natural examples and the robustness against Auto-Attack. For some methods, we also report the results of WideResNet-34-20. All results are from the original paper. From the table we can conclude that the robustness of both networks trained with CAT outperforms the previous methods, demonstrating the state-of-the-art performance obtained by our CAT.

Table 2: The white-box robustness results (accuracy (%)) of CAT on CIFAR-10 and CIFAR-100. We report the results of best checkpoint and last checkpoint. The best results are marked using **boldface**. Two ResNet-18 networks are used in our CAT framework. TRA-ALP is short for TRADES-ALP due to the limitation.

| Dataset | Method | Best Checkpoint | | | | | Last Checkpoint | | | | |
|---|---|---|---|---|---|---|---|---|---|---|---|
| | | Clean | FGSM | $PGD_{20}$ | $CW_\infty$ | AA | Clean | FGSM | $PGD_{20}$ | $CW_\infty$ | AA |
| CIFAR-10 | Natural | **94.65** | 19.26 | 0.0 | 0.0 | 0.0 | **94.65** | 19.26 | 0.0 | 0.0 | 0.0 |
| | AT | 82.82 | 57.57 | 51.76 | 50.05 | 47.55 | 84.53 | 53.90 | 43.56 | 44.19 | 41.57 |
| | TRADES | 83.17 | 59.22 | 52.63 | 50.79 | 49.21 | 83.04 | 57.46 | 49.81 | 49.01 | 47.03 |
| | ALP | 83.85 | 57.20 | 51.88 | 50.11 | 48.48 | 84.64 | 55.35 | 44.96 | 44.54 | 42.62 |
| | $CAT_{TRA-ALP}$ | 83.91 | **59.76** | **54.44** | 52.56 | **51.02** | 84.67 | **59.85** | 52.51 | 51.43 | 49.31 |
| | | 84.75 | **59.76** | 54.17 | **52.72** | 50.85 | 85.27 | 59.82 | **52.56** | **51.83** | **49.64** |
| CIFAR-100 | Natural | **75.55** | 9.48 | 0.0 | 0.0 | 0.0 | **75.39** | 9.57 | 0.0 | 0.0 | 0.0 |
| | AT | 57.42 | 31.90 | 28.78 | 27.27 | 24.88 | 57.34 | 26.77 | 21.24 | 21.50 | 19.59 |
| | TRADES | 56.98 | 31.72 | 29.04 | 25.30 | 24.23 | 55.08 | 30.40 | 26.81 | 24.78 | 23.68 |
| | ALP | 61.01 | 31.41 | 26.78 | 25.68 | 23.51 | 58.4 | 27.97 | 22.63 | 21.87 | 20.42 |
| | $CAT_{TRA-ALP}$ | 61.31 | 35.83 | **33.09** | 29.17 | **27.17** | 61.78 | **35.84** | **32.76** | **29.48** | **27.29** |
| | | 62.53 | **36.05** | 32.92 | 29.16 | 26.90 | 62.52 | 35.79 | 32.51 | 29.24 | 26.73 |

Table 3: The black-box robustness results (accuracy (%)) of CAT on CIFAR-10 and CIFAR-100. We only report the results of the best checkpoint. The best results are marked using **boldface**. Two ResNet-18 networks are used in our CAT framework. TRA-ALP is short for TRADES-ALP due to the limitation.

| Method | CIFAR-10 | | | | | CIFAR-100 | | | | |
|---|---|---|---|---|---|---|---|---|---|---|
| | FGSM | $PGD_{20}$ | $PGD_{40}$ | $CW_\infty$ | Square | FGSM | $PGD_{20}$ | $PGD_{40}$ | $CW_\infty$ | Square |
| AT | 64.54 | 61.70 | 61.57 | 61.42 | 56.16 | 39.15 | 37.56 | 37.46 | 38.85 | 30.11 |
| TRADES | 65.63 | 63.57 | 63.57 | 63.23 | 55.97 | 39.06 | 37.73 | 37.79 | 38.86 | 28.72 |
| ALP | 64.95 | 62.38 | 62.32 | 61.78 | 55.78 | 40.29 | 38.97 | 38.85 | 40.03 | 29.85 |
| $CAT_{TRA-ALP}$ | 65.73 | 63.65 | 63.78 | 63.24 | 57.55 | 42.26 | 40.76 | 40.76 | 41.78 | 33.04 |
| | **66.06** | **63.91** | **63.88** | **63.26** | **57.95** | **42.81** | **41.55** | **41.42** | **42.42** | **33.30** |

## 4.4 COMPARISION TO KD-AT

In general, the robustness of large models is higher than that of small models under the same training settings. For example, WideResNet-34-10 (Zagoruyko & Komodakis, 2016) trained by TRADES can achieve 53.08% robustness against AA, while the accuracy of ResNet-18 is only 49.21%. From this motivation, some authors have used knowledge distillation to distill the robustness of large models to small models and obtained good results. Considering that CAT also involves the collaborative training of two models, we compare CAT with the KD-AT method. To give a fair comparison, unlike the previous experiments using two same-size networks for CAT, we use two different-size networks for CAT training and then report the accuracy of both the two networks, i.e., WideResNet-34-10 and ResNet-18. Note that, unlike the KD method where the teacher is trained in advance, our CAT is trained with both the large model and the small model simultaneously, so there is no concept of teacher and student. In another word, we extend previous off-line distillation to an online way and achieve better performance.

Table 5 shows the results of Knowledge Distillation-AT and CAT, where ARD (Goldblum et al., 2020), IAD (Zhu et al., 2021), and RSLAD (Zi et al., 2021) are trained by KD-AT using TRDES trained WideResNet-34-10 network as teacher (second row in Table 5). CAT was collaboratively

Table 4: Quantitative comparison with the state-of-the-art adversarial traing methods. Two WideResNet-34-10 networks are used in our CAT framework.

| Method | Architecture | Nat | AA |
|---|---|---|---|
| Bag of Tricks for AT (Pang et al., 2020a) | WideResNet-34-10 | 86.28 | 53.84 |
| HE* (Pang et al., 2020b) | WideResNet-34-20 | 85.14 | 53.74 |
| Overfitting in AT* (Rice et al., 2020) | WideResNet-34-20 | 85.34 | 53.42 |
| Overfitting in AT (Rice et al., 2020) | WideResNet-34-10 | 85.18 | 53.14 |
| Self-Adaptive Training (Huang et al., 2020) | WideResNet-34-10 | 83.48 | 53.34 |
| FAT (Zhang et al., 2020) | WideResNet-34-10 | 84.52 | 53.51 |
| TRADES (Zhang et al., 2019) | WideResNet-34-10 | 84.92 | 53.08 |
| LLR (Qin et al., 2019) | WideResNet-40-8 | 86.28 | 52.84 |
| LBGAT+TRADES ($\alpha = 0$)* (Cui et al., 2021) | WideResNet-34-20 | **88.70** | 53.57 |
| LBGAT+TRADES ($\alpha = 0$) (Cui et al., 2021) | WideResNet-34-10 | 88.22 | 52.86 |
| LBGAT+TRADES ($\alpha = 6$) (Cui et al., 2021) | WideResNet-34-10 | 81.98 | 53.14 |
| LAS-AT (Jia et al., 2022) | WideResNet-34-10 | 86.23 | 53.58 |
| LAS-AWP (Jia et al., 2022) | WideResNet-34-10 | 87.74 | 55.52 |
| $\text{CAT}_{TRADES-ALP}$ | WideResNet-34-10 | 86.22 / 86.51 | 54.11 / **54.20** |
| $\text{CAT}_{TRADES-ALP}$ +AWP | WideResNet-34-10 | 86.74 / 87.01 | 56.43 / **56.61** |

Table 5: Quantitative comparison with the state-of-the-art Knowledge-Distillation AT methods. We use WideResNet-34-10 and ResNet-18 networks in the CAT framework for a fair comparison.

| Method | Architecture | Distillation | Role | Clean | AA |
|---|---|---|---|---|---|
| TRADES (Zhang et al., 2019) | WideResNet-34-10 | - | teacher | 84.92 | 53.08 |
| ARD (Goldblum et al., 2020) | ResNet-18 | ✓ | student | 83.93 | 49.19 |
| IAD (Zhu et al., 2021) | ResNet-18 | ✓ | student | 83.24 | 49.10 |
| RSLAD (Zi et al., 2021) | ResNet-18 | ✓ | student | 83.38 | 51.49 |
| $\text{CAT}_{TRADES-ALP}$ | WideResNet-34-10 | ✗ | - | **85.04** | **55.04** |
| | ResNet-18 | ✗ | - | 84.39 | 50.14 |

trained using two networks of different sizes. It can be seen that our method obtains good results and also obtains high clean accuracy.

## 5 CONCLUSIONS

In this paper, we first analyze the properties of different adversarial training methods and find that networks trained by different methods perform differently on sample instances, i.e., the network can correctly classify samples that are misclassified by other networks. Based on this observation, we propose a collaborative adversarial training framework, which aims to use the knowledge learned by peer networks to guide its learning. Thus improving the robustness of both networks. Extensive experiments on different datasets and different networks demonstrate the effectiveness of our approach, and state-of-the-art performance is achieved. Broadly, CAT can be easily extended to multiple networks for collaborative adversarial training, e.g, three peer networks. However, our method has some limitations the same as the previous method that adversarial training consumes training resources, and CAT has to train two networks simultaneously. Work on how to accelerate CAT will be carried out in the future.

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

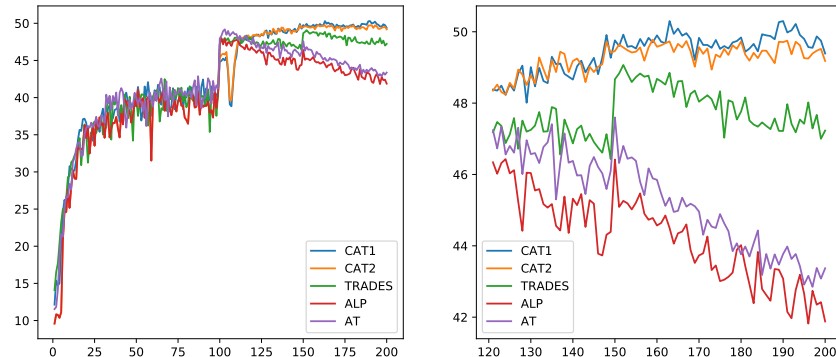

Figure 4: Test robust accuracy of AT, ALP, TRADES, and CAT with ResNet-18 on CIFAR-10 datasets. CAT can alleviate the problem of overfitting.

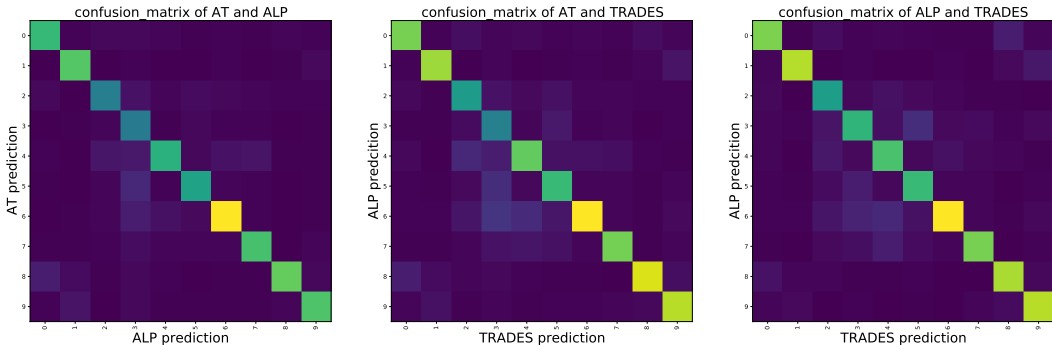

Figure 5: Confusion matrices of ALP-prediction and AT-prediction, TRADES-prediction and AT-prediction, ALP-prediction and TRADES-prediction with ResNet-18 on CIFAR-10 datasets. Confusion exists in different defenses methods.

## A  ALLEVIATE OVERFITTING

Overfitting in adversarial training is first proposed by Rice et al. (2020), which shows the test robustness decreases after peak robustness. And overfitting is one of the most concerning problems in adversarial training. Here, we investigated the overfitting problem in CAT. Results are illustrated in Fig. 4. Our CAT can alleviate the overfitting probalem that widely occurs in previous adversarial methods. Moreover, the performance for CAT has not saturated, and high performance is expected with longer epoch training.

## B  CONFUSION MATRIX

To better present motivation for CAT, we show the confusion matrices of different methods. In Figure 5, three confusion matrices are shown, i.e., ALP-prediction and AT-prediction, TRADES-prediction and AT-prediction, ALP-prediction and TRADES-prediction. Confusion exists in all three matrices, especially for blocks from class 3 to class 7. The conclusion is correspondence to Figure 1. The prediction intersection is reported in Tab. 6.

## C  CORRELATION BETWEEN DISCREPANCY AND CAT PERFORMANCE

In this section, we analyze the correlation between the discrepancy of different adversarial training methods and their adversarial robustness after CAT. First, we compute the prediction intersection

Table 6: The correlation between white-box robustness results (accuracy (%)) and prediction discrepancy of different CAT methods on CIFAR-10. Two ResNet-18 networks are used in our CAT framework.

| Method | PGD$_{20}$ | Prediction intersection |
|---|---|---|
| CAT$_{AT-ALP}$ | 53.11 | 81.02% |
| CAT$_{TRADES-ALP}$ | 54.44 | 78.95% |
| CAT$_{AT-TRADES}$ | 54.52 | 77.46% |

Table 7: The white-box robustness results (accuracy (%)) of CAT on CIFAR-10. We report the results of the best checkpoint. The best results are marked using **boldface**. **Two VGG-16** networks are used in our CAT framework. TRA-ALP is short for TRADES-ALP due to the limitation.

| Dataset | Method | Clean | FGSM | PGD$_{20}$ | CW$_\infty$ | AA |
|---|---|---|---|---|---|---|
| CIFAR-10 | AT | 78.31 | 53.11 | 48.39 | 46.32 | 43.69 |
| | TRADES | 79.11 | 53.75 | 48.28 | 45.93 | 44.63 |
| | ALP | 80.23 | 52.18 | 47.30 | 45.23 | 43.68 |
| | CAT$_{TRA-ALP}$ | 79.23 | 54.47 | **49.43** | 47.19 | **45.48** |
| | | 80.12 | **54.48** | 48.30 | **47.23** | 45.33 |

between different methods, formulated as:

$$intersection = \frac{1}{N} \sum_{x_i \in D} \mathbb{I}(f^{AT}(x_i), g^{TRADES}(x_i)), \tag{7}$$

where D is the datasets, and $\mathbb{I}$ is an indicator function, which is 1 when $f^{AT}(x_i) = g^{TRADES}(x_i)$ and 0 otherwise. The smaller this value is, the greater the discrepancy. Then, we report the adversarial robustness of CAT trained by different settings. Results are reported in Tab. 6. A conclusion can be drawn that the greater the discrepancy is, the higher the adversarial robustness after CAT.

# D    MORE EXPERIMENTAL RESULTS

## D.1    VGG-16 RESULTS ON CIFAR-10

The white-box robustness of VGG-16 (Simonyan & Zisserman, 2014) models trained using AT, ALP, TRADES, and CAT are reported in Tab. 7. The setting for VGG-16 is the same as ResNet-18 models, i.e., $\alpha = 1.0/20$ and $\beta = 1.0/20$. The improvement for CAT with VGG-16 models is as consistent with ResNet-18 models. CAT can boost model's robustness under AutoAttack with 2.0 points.

## D.2    MOBILENET RESULTS ON CIFAR-10

Similar to the above VGG-16 models, we report the while-box robustness of MobileNet (Howard et al., 2017) on CIFAR-10 datasets under various attacks in Tab. 8. The experiment set is the same as the previous setting. We can see that our CAT brings 1.0 improvement for MobileNet under AutoAttack, which is the most powerful adversarial attack method.

## D.3    RESNET-18 RESULTS ON TINY-IMAGENET

For the large-scale ImageNet dataset, just as all the baseline methods did not report the results, we are also unable to evaluate on ImageNet due to the very high training cost. To investigates the performance of our CAT in large datasets, we conduct the experiment of white-box robustness of

Table 8: The white-box robustness results (accuracy (%)) of CAT on CIFAR-10. We report the results of the best checkpoint. The best results are marked using **boldface**. **Two MobileNet** networks are used in our CAT framework. TRA-ALP is short for TRADES-ALP due to the limitation.

| Dataset | Method | Clean | FGSM | PGD$_{20}$ | CW$_\infty$ | AA |
|---------|--------|-------|------|------------|-------------|-----|
| CIFAR-10 | AT | 76.24 | 50.27 | 44.99 | 43.03 | 40.10 |
| | TRADES | 75.84 | 49.65 | 45.26 | 42.04 | 41.08 |
| | ALP | 79.46 | 50.14 | 43.95 | 42.08 | 40.01 |
| | CAT$_{TRA-ALP}$ | 80.14 | 51.25 | **46.38** | **44.24** | **42.20** |
| | | 79.86 | **51.28** | 46.22 | 44.05 | 42.16 |

Table 9: The white-box robustness results (accuracy (%)) of CAT on **Tiny-ImageNet**. We report the results of the best checkpoint. The best results are marked using **boldface**. Two ResNet-18 networks are used in our CAT framework. TRA-ALP is short for TRADES-ALP due to the limitation.

| Dataset | Method | Clean | PGD$_{50}$ | CW$_\infty$ | AA |
|---------|--------|-------|------------|-------------|-----|
| Tiny-Imagenet | AT | 43.98 | 19.98 | 17.60 | 13.78 |
| | TRADES | 39.16 | 15.74 | 12.92 | 12.32 |
| | ALP | 39.85 | 17.28 | 15.34 | 12.98 |
| | CAT$_{TRA-ALP}$ | 44.35 | 20.86 | 19.43 | 14.96 |
| | | **44.76** | **21.02** | **19.64** | **15.63** |

ResNet-18 on Tiny-ImageNet, which also is a widely used dataset in adversarial training. The results are shown in Tab. 9. Surprisingly, CAT shows impressive robustness on the large-scale dataset. The improvement is as significant as ResNet-18 in small datasets like CIFAR-10 and CIFAR-100.

## D.4 CAT OF ONE MODEL WITH VARIOUS ATTACKS

Table 10: The white-box robustness results (accuracy (%)) of CAT on CIFAR-10. We report the results of the best checkpoint. The best results are marked using **boldface**. PGD-CW denotes one network trained by PGD and CW. TRA-ALP is short for TRADES-ALP, denoting two networks with TRADES and ALP. TRA-TRA is short for TRADES-TRADES, denoting two networks with TRADES and TRADES. AT-ALP-TRA is short for ALP-ALP-TRADES.

| Dataset | Method | **Best Checkpoint** | | | | |
|---------|--------|-------|------|------------|-------------|-----|
| | | Clean | FGSM | PGD$_{20}$ | CW$_\infty$ | AA |
| | AT | 82.82 | 57.57 | 51.76 | 50.05 | 47.55 |
| | CAT$_{PGD-CW}$ | 82.09 | 56.48 | 52.48 | 49.28 | 48.06 |
| CIFAR-10 | CAT$_{TRA-ALP}$ | 83.91 | 59.76 | 54.44 | 52.56 | 51.02 |
| | | **84.75** | 59.76 | 54.17 | 52.72 | 50.85 |
| | CAT$_{TRA-TRA}$ | 81.94 | 58.85 | 54.19 | 51.52 | 50.30 |
| | | 82.13 | 58.77 | 54.02 | 51.56 | 50.14 |
| | CAT$_{AT-ALP-TRA}$ | 84.50 | 60.17 | 54.64 | 52.98 | 51.28 |
| | | 84.62 | **60.25** | 54.87 | 53.04 | 51.42 |
| | | 84.29 | 60.24 | **55.04** | **53.38** | **51.74** |

For our CAT method, we use two networks and two different attack methods for each network to perform adversarial training. An interesting baseline is one network with two different attack methods. Therefore, we use PGD and CW as our attack methods and one ResNet-18 as our network.

The results are reported in Tab. 10 (CAT$_{PGD-CW}$ entry). The improvement for this setting is not significant as the previous setting, but it still, boosts the model's robustness against all four attacks.

### D.5 CAT OF TWO MODELS WITH SAME METHODS

Another interesting baseline is two networks trained by the same adversarial training methods, i.e., two ResNet-18 networks are both trained by TRADES. We denote this setting as CAT$_{TRADES-TRADES}$. The results are reported in Tab. 10. The improvement for this setting is not significant as the previous setting, but it still, boosts the model's robustness against all four attacks. However, the improvement is more significant than just using one network. A conclusion can be drawn that two networks are important for CAT to achieve better adversarial robustness.

### D.6 CAT OF THREE MODELS WITH THREE METHODS

Our CAT is a generalized method, which can use any number of different adversarial training methods for collaborative learning. Here, we report the results of CAT trained with three adversarial training methods in Tab. 10 (CAT$_{AT-ALP-TRADES}$). The robustness improvement is more significant than CAT trained with two adversarial trained methods, which shows the generalizability of our CAT.

