# OpenReview forum: "CAT: Collaborative Adversarial Training"
_ICLR.cc/2023/Conference — Submitted to ICLR 2023_

### Official Review · Reviewer_JNu7 · 2022-10-28

**Confidence:** 4
**Correctness:** 3
**Technical Novelty And Significance:** 3
**Empirical Novelty And Significance:** 3
**Recommendation:** 5

**Clarity, Quality, Novelty And Reproducibility:**

The paper is clear and well-written. Hyperparameters and other training settings have been discussed. The authors are encouraged to make the code available too.

**Strength And Weaknesses:**

Strengths-

- The proposed approach is intuitive and simple to implement.
- The work shows improved robustness compared to the baseline models trained non-collaboratively.
- The paper is clear and well-written.

Weaknesses/ Questions -

- The proposed method involves higher complexity when compared to baselines - 2x computational cost, 2x memory overhead, more hyperparamaters.
- Could the authors elaborate further on why and how CAT improves performance? How are these adversarial training methods different in the first place - and why does combining them result in improvements?
- Could the authors share the confusion matrix of (correct and incorrect) predictions from TRADES and PGD-AT (or ALP)? This would better motivate the approach.
- The best results on Robustbench are actually obtained by applying Adversarial Weight perturbation (AWP). Could the proposed method TRADES-ALP also be integrated with AWP? It would be useful to compare gains of AWP and no-AWP runs.
- Could the authors share an ablation of TRADES-TRADES, where the same method (TRADES) is used for training both models?




**Summary Of The Paper:**

The paper proposes a Collaborative Adversarial Training (CAT) framework, which is based on the observation that robust models trained with different training methods have different performance on different instances, despite the models having similar accuracies. This gives rise to the proposed collaborative approach: Given the setup of two networks being trained with two different training methods, an adversarial sample generated by one network is fed to the other network to obtain the corresponding logit, which is then utilized to guide the learning of the first network. The evaluation is performed on two benchmark datasets CIFAR10 and CIFAR100 against a number of baselines to test the effectiveness of the approach in improving model robustness. The authors also compare their method to various knowledge distillation techniques which work on a similar principle.

**Summary Of The Review:**

This work proposes a training regime to combine the benefits of different adversarial training algorithms, which is interesting and novel. However, there is little motivation on why this works, and how the trained models are different. Including some important ablations (mentioned above) can improve clarity on the impact of the proposed method.

---

### Official Review · Reviewer_df3y · 2022-10-29

**Confidence:** 4
**Correctness:** 2
**Technical Novelty And Significance:** 2
**Empirical Novelty And Significance:** 2
**Recommendation:** 3

**Clarity, Quality, Novelty And Reproducibility:**

The clarity is good.

Quality and novelty are low.

The reproducibility is unknown.

**Strength And Weaknesses:**

Pros.

1. The paper is well-written and easy to follow.
2. Robustness is evaluated by diverse adversarial attackers beyond PGD.



Cons.

1. The collaborative training technique can be regarded as a "soft data augmentation" since it will use the input sample and associated predictions from the peer networks. Therefore, (a) the authors need to compare with data augmentation approaches in adversarial training like "Data Augmentation Can Improve Robustness"; (b) it is hard to say that the comparison in this paper is fair since CAT use at least 2 times resource cost than other approaches.
2. Only small-scale datasets are considered like CIFAR-10 and CIFAR-100. It is insufficient to support the effectiveness of the proposed methods. Larger datasets like ImageNet are required.
3. Meanwhile, the network backbones for evaluation are also limited. More architectures like VGG and MobileNet are needed.
4. The provided method is motivated by the prediction discrepancy between different robustified networks. However, it is unconvincing between several important baselines are missing. For example, the prediction intersection results for the same AT method with different random seeds or different random start, are required. Whether the prediction difference between TRADES and AT is larger than the results between two AT runs?
5. More ablations are needed. For instance, how about involving more attackers (e.g., FAT, CW, auto attack) in collaborative training?

**Summary Of The Paper:**

Summary.

This paper is dedicated to developing improved adversarial training methods. The authors are inspired by the observations of prediction discrepancy from different adversarially trained models. Specifically, they introduce a collaborative adversarial training framework (called CAT) to improve the robustness. CAT inputs the adversarial example generated by each network to the peer network and uses the peer networks' logit to guide its training. Experiments are conducted on CIFAR-10 and CIFAR-100.

**Summary Of The Review:**

Incremental ideas with insufficient studies.

---

### Official Review · Reviewer_a6Jb · 2022-10-29

**Confidence:** 4
**Clarity, Quality, Novelty And Reproducibility:** Pls see the above comments.
**Correctness:** 2
**Technical Novelty And Significance:** 2
**Empirical Novelty And Significance:** 2
**Recommendation:** 3

**Strength And Weaknesses:**

Strength:

1. The idea is straightforward, and the paper writing is easy to follow, which simply ensembles the knowledge from different adversarial training models.

2. The experiments are extensive among a variety of datasets and networks, and the evaluation achieves state-of-the-art results.


Weaknesses:

1. The idea of the proposed method is quite lacks novelty. Similar ideas of collaboratively fusing knowledge have been proposed and studied for improving robustness [1][2][3].

2. The paper does not theoretically analyse the benefits brought by fusing different adversarial training methods. The trivial experimental results do not clearly support their claims, the author seems to simply blend them together to get improvement without any further analysis.

3. Is that necessary to train two different robust neural networks from scratch? Efficiency is the main challenge in adversarial training, and now the authors propose to scarify by doubling the time, making it worse.

4. What if we train one neural network and fuse the knowledge by using different adversarial examples generated by different methods? This is a strong baseline that should be concluded for comparison.

5. What is ALP? The abbreviation occurs suddenly in the experiment part without any reference or explanation.

[1] Toward Learning Robust and Invariant Representations with Alignment Regularization and Data Augmentation

[2] Improving adversarial robustness by learning shared information | Elsevier Enhanced Reader

[3] Prior-Guided Adversarial Initialization for Fast Adversarial Training

**Summary Of The Paper:**

The authors observe a phenomenon that robust models adversarial trained by different methods (like AT and TRADES) have different reactions to the given input. Motivated by this, they proposed to perform collaboratively adversarial training to improve the robustness by training two neural networks from scratch.


**Summary Of The Review:**

Pls see the above comments.

---

### Official Review · Reviewer_MzNC · 2022-11-02

**Confidence:** 4
**Correctness:** 4
**Technical Novelty And Significance:** 3
**Empirical Novelty And Significance:** 3
**Recommendation:** 5

**Clarity, Quality, Novelty And Reproducibility:**

The paper is easy to understand and follow. The authors should proofread the paper to remove some of the typos (e.g. "the TARDES-trained network). The idea of combining multiple defenses is not novel. There are several methods for training an ensemble of adversarially-trained models to improve robustness [1, 2], which were not cited nor compared with. The authors should relate the proposed method to ensemble methods for adversarial training. In comparison, the authors should consider using some of the more recent and stronger baselines for comparison.

[1] Tramèr, F., Kurakin, A., Papernot, N., Goodfellow, I., Boneh, D., & McDaniel, P. (2018). Ensemble adversarial training: Attacks and defenses. In , International Conference on Learning Representations (pp. ). : .
[2] Pang, T., Xu, K., Du, C., Chen, N., & Zhu, J. (2019). Improving adversarial robustness via promoting ensemble diversity. In K. Chaudhuri, & R. Salakhutdinov, Proceedings of the 36th International Conference on Machine Learning (pp. 4970–4979). Long Beach, California, USA: PMLR.

**Strength And Weaknesses:**

### Strengths

- Proposed a framework for collaborative adversarial training, where two robust models are trained jointly.
- Demonstrated promising experimental results on CIFAR-10 and CIFAR-100 benchmarks.

### Weaknesses

- The increased computational cost of training two robust models.
- The lack of a theoretical analysis of the proposed method. It is not clear why the proposed objective will guide the training toward a more robust model compared to the single model, single attack, and single objective training.
- The lack of more detailed experimental analysis and other baselines for comparison.

### Questions
- Can collaborative adversarial training be affected by catastrophic/robust overfitting?
- Does collaborative adversarial training perform better than training one robust model with multiple attacks and/or multiple weighted objectives?
- Can you include experiments with more than 2 defenses?

**Summary Of The Paper:**

The authors made an observation that different adversarial defense strategies make different mistakes. Based on this observation, they proposed collaborative adversarial training, where simultaneously train two robust models. The objective of collaborative adversarial training is to minimize the symmetric KL-divergence between the logits of the first and second models. In the experiments, the authors showed improved robustness against AutoAttack on CIFAR-10 and CIFAR-100 datasets.

**Summary Of The Review:**

The authors proposed a collaborative adversarial training method that combines two defense methods. The approach is interesting and marginally novel. It improves robustness upon TRADES and ALP baselines. However, the paper contains many typos, the empirical comparison does not include more recent SOTA methods, and some citations (such as ensemble adversarial training) are missing.

---

### Decision · Program_Chairs · 2023-01-20

**Decision:**

Reject

**Justification For Why Not Higher Score:**

This paper received 4 reviews, all the reviewers have recommended rejection of the paper. After rebuttal, the scores are still not changed. Therefore, there is no reason to accept the paper.

**Justification For Why Not Lower Score:**

N/A

**Metareview: Summary, Strengths And Weaknesses:**

This paper proposes a Collaborative Adversarial Training (CAT) framework that combines the benefits of different adversarial training algorithms, based on the observation that robust models trained with different methods have different performance on different instances, despite the models having similar accuracies.

After rebuttal, the paper receives scores of 3355. The authors have done a good job during rebuttal, and added additional results and analysis. However, several concerns still remain. (1) Multiple reviewers mentioned that the proposed method involves higher complexity (at least 2 times resource cost) than baselines, and it remains unclear whether such comparison is fair or not. (2) A theoretical analysis of the proposed method is lacking, thus also making the proposed method not sufficiently motivated. It remains unclear why the method works, and how the trained models are different. (3) Only small-scale datasets are considered like CIFAR-10 and CIFAR-100. Larger datasets like ImageNet are required.

The rebuttal is unfortunately not convincing enough for reviewers to increase the scores. Therefore, the AC would like to recommend rejection of the paper.